# Exploring Muslims’ Health-Related Behaviours in Portugal: Any Impact on Quotidian Community Pharmacy Practice?

**DOI:** 10.3390/pharmacy10030055

**Published:** 2022-05-20

**Authors:** Aisha Omar, Grishma Dramce, Dragana Lakic, Afonso Cavaco

**Affiliations:** 1Department of Pharmacy, Pharmacology and Health Technologies (Social Pharmacy), Faculty of Pharmacy, University of Lisbon, Av. Prof. Gama Pinto, 1640-003 Lisboa, Portugal; aishaomar@campus.ul.pt (A.O.); grishmad@campus.ul.pt (G.D.); 2Department of Social Pharmacy and Pharmacy Legislation, Faculty of Pharmacy, University of Belgrade, Vojvode Stepe 450, 11221 Belgrade, Serbia; dragana.lakic@pharmacy.bg.ac.rs

**Keywords:** community pharmacy, inter-cultural practice, Islamic medicine, Muslims, Portugal

## Abstract

Muslims are a growing community in European countries. General health habits, including therapy-related behaviours, have been described, though implications to pharmacy practice might vary with the local dominant culture and setting. This exploratory study aimed to describe Muslims’ prevalent health and medication-related practices and possible implications for culturally competent community pharmacy practice. A descriptive cross-sectional survey was administered to a convenient sample of 100 participants at Lisbon Central Mosque, Portugal. Demographics, dietary, Traditional Arabic and Islamic Medicine (TAIM) and religious practices were examined, including health conditions and conventional biomedical treatments. Participant reported ailments (26%) were aligned with prevalent conditions in the general population. Ill participants were significantly associated with TAIM and Islamic dictates (*p* < 0.05), particularly Zam-Zam water and milk thistle usage. Participants’ orientation to dietary options and Qur’an restrictions were observed regarding forbidden substances in medication, raising issues on medication adherence for some oral dosage forms. TAIM and religious beliefs supplement illness recovery and health improvement instead of replacing conventional healthcare in a religious minority well integrated within the dominant culture. Portuguese community pharmacists should not neglect religious specificities if seamless care is delivered, enhancing professionals’ collaboration skills with multicultural patients.

## 1. Introduction

Muslims in western European countries are a growing religious community. According to the Pew Forum, an estimated 19 million Muslims live in the European Union (3.8%) [1]. Portugal, a small southwestern European country, is no exception to Muslim presence. Historical Islamic roots to migration from ex-colonies such as Mozambique and Guinea-Bissau (in the 1970s) [2] were supplemented in the last decades with Muslims arriving from the north of Africa, Bangladesh, Pakistan, and some Arab countries [2,3]. Presently, amongst nearly 10 million inhabitants, mainly of the Roman Catholic faith [4], there are approximately 50,000 Muslims, of which almost two-thirds reside in the Greater Lisbon area [5].

As with other faiths, Muslims bring their religious and social traditions, practices, and health beliefs to daily life, including disease treatment and medication use [6]. For instance, the Islamic conceptions of modesty and privacy and the misconceptions in illness aetiology contribute to an increased health risk by regular healthcare services avoidance. Dietary options such as halal food, integral to the healing process, may also impact health [7,8]. In addition, reported delays in care-seeking due to a perceived lack of female Islamic clinicians happen in women’s health [9,10].

Compared to Western medicine, health and care differences can be derived from traditional Arabic and Islamic medicine (TAIM) principles. One TAIM difference comprises prescribed medication usage. For instance, Muslim patients may show concerns about taking drugs such as sedative medications since the ethical-legal tradition discourages the use of mind-altering substances [8]. Furthermore, medications from an animal origin (e.g., pig) are forbidden and may be avoided [8], particularly while observing religious fasts [11,12]. Treatment adherence can be compromised, and promoting adherence is an important duty for community pharmacists, amongst several others [13,14,15]. Pharmacists are globally concerned with a seamless practice, including intercultural competencies, cultural awareness, and sensitivity [16]. For instance, The American College of Clinical Pharmacy (ACCP) has issued documents discussing the effective delivery of culturally and linguistically appropriate pharmacy services in cross-cultural settings, including education and policy procedures [17,18]. Additionally, U.S. pharmacy schools tend to include lectures regarding therapeutic management in special populations, including Muslims, and train pharmacy students to acquire cultural competency.

The Portuguese pharmacy professionals and educators have had limited access to national or local information regarding cultural minorities’ health beliefs and behaviours, notably therapeutic and medication preferences. This topic is generally absent from undergraduate curricula and continuous or vocational training of Portuguese pharmacists. No previous studies regarding Portuguese Muslim patients, their medication, and pharmacists’ role are known. This study aimed to obtain a culturally situated description of the Muslim healing attitudes, particularly those related to medication usage, rich enough to endorse pharmacy students’ and professionals’ intercultural practice needs.

## 2. Materials and Methods

The present quantitative exploratory study followed a cross-sectional descriptive design. After informed consent, a survey was conducted to assess health and medication-related behaviours based on anonymous paper-based questionnaires handed to individuals who prayed in the Lisbon Central Mosque.

### 2.1. Traditional Arabic and Islamic Medicine

The conceptual model of TAIM proposed by Alrawi and Fetters (2012) includes the influences of traditional healing systems, such as the Ayurvedic and Traditional Chinese Medicine (TCM) [19]. As is widely known, Muslim health beliefs and behaviours are also deeply rooted in prophecies and sacred texts. These features informed the development of the survey, although no further aspects regarding the healing system were aimed at in the present study.

According to Alrawi and Fetters (2012), there are three main approaches to therapeutic plans in TAIM, originating or consequential to the Islamic prophetic tradition, including the use of texts such as those found in the Qur’an and the Sunnah (Figure 1). Spiritual healing is the shared element to which the believer adds: (A) dietary practices such as fasting and drinking Zamzam water, (B) mind-body therapy, which includes prayer and chanting, and (C) applied therapy, which includes massage and cupping. Medicinal herbs are also part of the Muslim healing options.

#### 2.1.1. Dietary Practices (A)

Muslim dietary practices use various functional foods (or nutraceuticals) prescribed by the Islamic prophetic traditions [20,21,22]. One example is Nigella sativa, also known as black cumin (in Portuguese), black seed (in English), or kalonji (in Arabic). Nigella seeds, which are the active part of this plant [23,24], present healing properties according to the Hadith (i.e., the sayings of the Prophet). Scientific studies have confirmed the high thymoquinone (TQ) levels, a biologically active substance in the volatile oil extracted from the seeds [23]. In addition, TQ has shown immune-stimulatory, gastroprotective, hepatoprotective, nephroprotective, and neuroprotective activities [25].

The oral tradition in the Portuguese Islamic community reveals the most used foods presenting healing properties to be honey, olive oil, and fruits such as dates and figs. Honey is believed to improve circulation, relieve intestinal colic, and be a topical antibiotic [26,27]. Olive oil is well known for its beneficial effects on cardiovascular and metabolic disorders through its phenolic components [22,28]. The religious indication is to expel renal stones [29]. Dates are a fruit widely used for healthy eating, as a source of energy and micronutrients, also interesting for several diseases, including heart conditions and diabetes [22,30]. Besides being a nutritious fruit, figs have anti-cancer properties and benefit the gastrointestinal tract, respiratory tract, and cardiovascular disorders. At the same time, religious indications comprise treatment of swelling, pharynx issues, and diuretic actions [29].

Other plants described in the Holy Qur’an as having medicinal properties, such as cucumber, garlic, ginger, lentils, onion, and pumpkin, among others [22], are usual cooking ingredients in Mediterranean cuisine. These were left to participants’ voluntary mentioning in the questionnaire. The questionnaire also comprised the most indispensable nutrient, water. It is believed amongst the Portuguese Islamic population that the holy Zamzam water has healing properties. The source is in the city of Makkah, Saudi Arabia. Zamzam water presents alkaline and antioxidant properties, rich in several vital minerals [31,32].

#### 2.1.2. Mind and Body Therapy (B)

TAIM mind-body therapy also encompasses spiritual healing. It is part of the Islamic tradition in Portugal to associate the recitation of prayers (chants) and verses from the Qur’an, recommended by the Prophet Muhammad [33]. The Qur’an verses of healing (syifa) are essentially six: 9:14, 10:57, 16:69, 17:82, 26:80, and 41:44. In addition to these verses, there are other Qur’an passages also advised by the Prophet to be recited for general protection and invocation of help from God. For example, the recitation of 1:1 “The Opening”, known as the longest chapter in the Qur’an, is mandatory in the five daily prayers and helps cure diseases. The 36:58 “Ya-Sin,” also called the light of the Qur’an, is advised to be recited every day at dawn and dusk for protection from any disease, amongst several others [34]. Verses were included in the questionnaire to measure religious devotion and assess relationships with treatment behaviours.

#### 2.1.3. Applied Therapy (C)

TAIM also includes massage, hydrotherapy, and cupping [19]. Cupping is a complementary and alternative medicine (CAM) originally from China. This therapy is believed to correct imbalances in the internal biofield by restoring the flow of Qi (“life force” in Chinese). It involves placing heated cups on the skin to create suction to facilitate healing by mobilising the blood flow [35]. These items were included in the survey questionnaire to explore the healing habits of the Lisbon Muslim community.

TAIM followers also use medicinal plants, as in the majority of the Portuguese population. Two frequent examples are *Matricaria chamomilla* L. (golden chamomile) and Silybum marianum (milk thistle) [36]. The golden chamomile’s biological activity is primarily due to its flavonoids and essential oils, delivering anti-inflammatory, antiseptic, and sedative properties. One of the most known of several golden chamomile indications is the treatment of sleep problems [37]. The milk thistle is mainly used for liver and biliary diseases. Its active substance (silymarin) is a mixture of flavonoid complexes that protect the liver and kidney cells with anti-cancer effects [38]. These two plants were specified in the questionnaire.

### 2.2. Survey Instrument

The questionnaire in Portuguese was carefully designed from the study outset and comprised of three main sections. The first covered participants’ sociodemographic data (10 questions), including religious habits, such as the degree of compliance with the dictates of the religion, following the precepts of the Qur’an and Sunnah, and the practice of religious fasts in the month of Ramadan. Independent numeric rating scales with (1) never, (2) sometimes, and (3) always were used for this purpose. The second section contained 15 open-ended questions about health and disease status, conventional medicine and/or CAM use, prescribed medications, and perceived limitations to conventional treatments emerging from cultural and religious boundaries. The third section contained six dichotomous response choice questions regarding the use of honey, olive oil, black cumin, Zamzam water, golden chamomile, and thistle while including an open field for other products. The questionnaire asked about the purpose and how it was used for all products, and verbatim replies were content analysed, coding those textual answers into meaningful categories for further quantitative analysis of dichotomous variables [39]. Coding doubts were triangulated among the research team members, and any discrepancies were solved by consultation with a Muslim community leader.

The survey face validity was initially established by faculty experts browsing for error checking (e.g., double-barrelled, confusing, or leading questions). Next, the Lisbon Islamic community leaders were invited to comment on the questionnaire scope readability; adjustments were made accordingly. The questionnaire was piloted on a subset of 10 individuals from the population of interest. No significant issues were found, and the survey was considered ready to be conducted. No further validation, including psychometric properties, was aimed.

The study’s ethical clearance was granted by the Lisbon Islamic Community Council (the equivalent of Institutional Review Board Comunidade Islâmica de Lisboa—Mesquita Central), chaired by the spiritual leader and legal guardian of the community (Imam Sheik Munir). The Imam’s involvement also improved study participation and response. Furthermore, the study followed all good practices in healthcare investigation, including data management and storage. The questionnaire was entirely anonymous and voluntary, and participants signed an informed consent form prior to data collection.

### 2.3. Sample and Sampling Procedures

The exploratory design warrants a convenient, non-probabilistic sampling procedure, using a personal invitation from a field researcher at the Lisbon Central Mosque. Lisbon’s metropolitan area has the largest Muslim community in the country. Based on regular service attendance (at least once a week), participation occurred between 15 March 2019, and 30 September 2019, based on expected service attendance (at least once a week). Other participant inclusion criteria comprised 18 or more years of age and self-declared proficiency in written Portuguese.

### 2.4. Data Collection and Analysis

During questionnaire administration, the field researcher was always present and resolved any potential questions without undue influences. The leading investigator irreversibly coded the answers and saved them in an MS Excel file on a computer with reserved access. Afterwards, data were imported into the IBM SPSS software (v26, IBM SPSS, Chicago, IL) for statistical analysis. Descriptive methods were used (frequencies and medians), as well as non-parametric tests to identify differences between groups, knowing data were not normally distributed [40]. The Mann–Whitney (U) and Kruskal–Wallis (H) tested differences between means of independent samples. All statistics were used with the Type I (α) probability of 0.05.

## 3. Results

### 3.1. Sociodemographic and Religious Data

One hundred Muslim individuals replied to the survey. The sample’s median age was 27.5 years (IQR = 19), with 51 being male. Participants’ education ranged from elementary to higher education, with most of the sample having a higher education degree (minimum three-year Bachelor) (*n* = 44) and secondary schooling (*n* = 26). Fifty-nine participants were married, and 49 declared a household income above 1000 EUR per month in family life. In terms of religious practice, the sample observed the fundamental principles of following the Islamic dictates and Qur’an (median 2.4, IQR = 1), while fasting had the highest possible value (median 3, IQR = 1).

### 3.2. Health Data and Conventional Healthcare

Participants’ reported health status found 63 participants with no conditions, while eight mentioned two or more ailments. Twenty-two different chronic diseases (according to the ICD v10 classification) were declared, with the three most common being from the cardiovascular system (mainly hypertension and hypercholesteremia, *n* = 12), diseases of the respiratory system (*n* = 9), and disorders of the digestive and the immune system (both with *n* = 7). All participants who reported a health problem affirmed taking at least one pharmacy medication. Painkillers were the most frequently reported (*n* = 15, mainly paracetamol), followed by cardiovascular medication (*n* = 14) and antiallergic drugs (*n* = 11) (Table 1).

### 3.3. Complementary and Alternative Medicines

At least 26 participants used CAM to treat their illnesses, with cupping (8) and acupuncture (5) the most frequent TCM option. Six participants reported only TAIM, and five mentioned homoeopathy, while at least three participants simultaneously used more than one CAM option. There were no significant associations between CAM and participants’ demographics, such as age, gender, or income. Those self-reporting an ailment were statistically associated with the use of CAM options, such as TCM (U = 1013.5, *p* = 0.023), TAIM (U = 1026.4, *p* = 0.016), and Ayurvedic (U = 1039.5, *p* = 0.008). No CAM options had a statistical association with Islamic religious behaviours.

### 3.4. Religion and Healthcare

Eighty-nine participants reported reciting verses from the Qur’an that relate to health protection and healing (1:1, 2:255, 1:285, 1:1286, 9:14, 10:57,16:69, 17:82, 26:80, 36:58, 41:44, 55, 112, 113, and 114). Approximately two-fifths of the sample (41%) always recite verses when suffering from a health condition, in particular verses 1:1 (34%) and 2:255 (25%), on their own or in conjunction with other verses, both during prayers and on other occasions. Sixty participants associate the recitation of verses with conventional medical treatment, including medication intake, while 49 associate the verses when using CAM options. No statistical association was found between reading the Qur’an and the use of CAM, but that was not the case for participants who followed Islamic dictates (U = 83.5, *p* = 0.015). Following religious dictates and fasting were not related to adherence to chronic drug treatment, although 15 participants took their medications before sunrise and after sunset in Ramadan.

Participants were asked whether religion would interfere with conventional medical treatments; 26.2% admitted some degree of interference based on religious principles observation, although no associations were found between participants’ reported adherence to medical treatments and religious variables, such as praying and fasting. On the other hand, the Qur’an restrictions were observed regarding forbidden substances. Sixty-two reported not taking gelatine-based products, and 83 would not take a medication that might have alcohol in its composition. This behaviour confirmed a significant association between strict fasting followers and not taking encapsulated drugs (U = 338.5, *p* = 0.019). The Qur’an commandment not to fast in case of chronic diseases was followed by 67.2%, but not statistically associated with CAM options, including TAIM.

### 3.5. TAIM and Healthcare Products Usage

Table 2 presents medicinal plants and functional foods and their everyday use for the study participants. While some participants mentioned specific or targeted healthcare use, most were described for nonspecific healing, i.e., helping protect good health or prevent ill-health. The most frequently used product was honey for helping with upper respiratory tract symptoms (e.g., sore throat), followed by olive oil and Zamzam water, both contributing to good general health.

Other plants were mentioned as being consumed based on their believed positive impact on health. Those were beetroot (4%), cucumber (4%), garlic (3%), ginger (3%), barley (2%), olives (2%), pumpkin (2%), moringa (2%), turmeric (2%), and vinegar (1%).

A statistical association was found between TAIM health products usage and socio-demographics for Zamzam water. Being a female (U = 920.5, *p* = 0.008), married (H = 9.192, *p* = 0.027), and having a higher income (H = 9.880, *p* = 0.007) was associated with higher consumption. The consumption of milk thistle was also significantly related to married participants (H = 18.672, *p* < 0.001) and to those fasting (U = 585.5, *p* = 0.032). Regarding associations between TAIM health products and being ill, the use of black cumin seeds presented a significantly higher proportion than those not suffering from a condition (U = 885.5, *p* = 0.004).

## 4. Discussion

Pharmacy practice and the community pharmacists’ role have faced many challenges worldwide, and Portugal is no exception [41]. One challenge is seamless and inclusive pharmaceutical care, particularly relevant for multicultural and religious minorities [42,43]. This applies to the growing Islamic community in urban areas [44]. Therefore, Muslims’ health behaviours, mainly the habits influencing the interplay between religious traits, the alternative treatments of diseases, and conventional medicine, were explored in this study.

The study sample reflected a relatively young group of participants, although no available data exist on the basic demographics of the Muslim population from Lisbon or Portugal. The sample level of schooling (44% with higher education) was higher when compared with the 23.6% of the general Portuguese population (INE—National Statistics Institute, 2019 data). Both younger and educated participants might result from reaching participants in the Central Mosque, located in an inner Lisbon area, where firms and tertiary services operate, and fewer households exist, including Muslim residents. Nevertheless, half of the sample’s income was below the INE declared average gross monthly salary in Portugal (1314 EUR, 2020 data). The sample was above the 47% of the married population in Portugal, with almost nil divorced (1%) participants, suggesting that most were following the spiritual encouragement to constitute a family [45]. Mean age, education level, income and civil status reflect a sample that might differ from Portugal’s overall Muslim population demographics.

Sixty-three per cent of the sample reported not suffering from any disease, and the relatively young participants may explain this finding. The cardiovascular system prevails for those with a health condition, like the general Portuguese population [46]. This alignment with the general population should be favourable for ensuring the equivalent and positive community pharmacists’ contribution to patients’ treatment outcomes [47,48].

The disease is ascribed to limited interest in human beings’ spiritual elements in Islamic culture. Herbal remedies should be added to faith-healing through prayer and the recitation of holy verses [49]. Interestingly, participants do not follow the rule of reciting the six verses mentioned in the Holy Qur’an specifically indicated for healing [34]. Qur’anic medicine prescribes disease treatment by reading or listening to verses from the Qur’an. The absence of this procedure in our sample may be due to no Qur’anic medicine centres and few Islamic schools or madrasas in Lisbon. Still, participants’ compliance with religious traditions and the commandants of the holy Qur’an was observed, as well as the extensive fasting. These are to happen simultaneously with conventional or alternative treatments, as encouraged by the Qur’an and Sunnah, the body of traditional social and legal custom and practice of the Islamic Community [34]. Concomitant praying or recitation and medical treatments prescribed by healthcare professionals’ do not harm the prescribed treatments and may benefit from placebo effects. There are no conflicts in the Arab world between Western medicine and the Islamic faith [49]. Hence, pharmacists should be prepared to accept the firm belief in the spiritual component of the healing process, like other faiths [50]; for instance, many Catholic believers in Portugal pay off their promises to saints and the church after being cured of severe disease. Knowing that Muslim patients see health professionals as assistants to God’s will [51], pharmacists should assume their role as part of the healing process, paying attention to medication features, including the Halal requirements of health products [51,52].

One of the most used CAM in this sample was cupping. This technique, highly recommended by the Prophet Muhammad, is also a frequent healing complement in countries with well-established conventional medicine systems, such as Germany, Norway, and Denmark [53]. Other CAM options were also chosen by those presenting ill-health, but no associations with religious behaviours were found. This has two implications: conventional medicine has an essential role in solving Muslims’ health problems, and there is a choice for CAM beyond prayers and dictates. Thus, pharmacists should not ignore having Islamic patients using medicinal herbs and other TAIM health products.

The three most used TAIM products for therapeutic purposes were honey, Zamzam water, and olive oil. Honey is a product widely consumed by Muslims in general for the belief in its healing power; actually, the Qur’an has a chapter exclusively referring to bees [34]. However, the community pharmacist must inform that honey consumption may be harmful to, e.g., diabetic patients [51]. Olive oil is an ingredient widely used in the Mediterranean diet and, coincidentally, considered very suitable for treating (and preventing), e.g., cardiovascular diseases [51]. More interesting is the use of Zamzam water. For instance, it is believed to heal sickness and relieve medication’s side effects in cancer patients [54,55]. Zamzam’s ad-hoc descriptions disclose the belief that Zamzam has synergistic effects over the medication, reducing the dosage, treatment, and duration of treatments. Summed to women’s role in family care, paediatric medicines may affect the care provided.

Milk thistle is a cultivated edible plant not mentioned in the holy texts. However, it has belonged to the Greco-Arab and Islamic medicine as a liver protector, tonic, and natural therapeutic agent [36]. Participants reported its use for digestive health problems, with a significant association found with fasting and strict control of food intake, reinforcing the attention to the digestive system. Regarding black cumin seeds, no significant associations were found with a specific health condition, strengthening its use according to the religious literature for overall health benefits and healing properties [23,36]. No relevant drug–drug interactions were found with conventional drugs for thistle, cumin, or even chamomile [56].

In Portugal, a predominantly Catholic country, more inadequate knowledge of Halal pharmaceuticals is expected to exist than in Muslim countries [57,58]. Avoiding certain medications because they may contain prohibited substances (e.g., those having a porcine origin) can explain why more than half of the participants said they do not take medications containing gelatine in their composition if unable to confirm the medication ingredients from the information or package leaflet, which raises medication adherence issues. The pharmacists’ role is to be alerted to this possibility and provide information to assure the medications are free of prohibited products or inform viable alternatives. These include products manufactured using Halal bovine or fish gelatine [59]. Non-adherence to treatments can also happen with liquid medications perceived as containing alcohol in their composition [58].

Finally, the culturally competent pharmacist must consider the impact of fasting, such as in Ramadan. Although Muslim patients are exempted from fasting if there is a chronic disease, one third of the sample comply with fasting rules and might adjust the medication schedule and doses. Pharmacists’ communication with patients for medication adjustments has shown significant gaps [60], particularly critical for diabetic patients [61,62,63]. Portuguese pharmacists involved in the healthcare of cultural minorities can work to adjust the therapy and contribute to the education of Muslim patients and other cultural minorities [43,62].

### Study Strengths and Limitations

The study’s main strength relies on its contribution to Portugal’s intercultural pharmacy practice. According to the authors’ best knowledge, this is the first study to address practical aspects of Muslim patients’ medication use in Portugal and how religious features might influence the pharmacists’ work.

Study limitations are related to TAIM as an extensive area of study, permeated with social and cultural factors not addressed in this study, thus limiting the scope and depth of evidence interpretation. In addition, no statistical representation of the community is guaranteed, although the community spiritual leaders did not recommend extending questionnaire participation. Furthermore, the study sampling was affected by self-selection bias, with a proportion of higher educated younger participants, which indicates further caution in extending the findings to the overall Portuguese Muslim population. Older generations usually tend to depend more heavily on CAM, traditional medicine, and prayers than younger and more Westernised patients.

## 5. Conclusions

The preliminary findings of this study showed a population that follows the Holy Qur’an’s emphasis on a healthy lifestyle, including functional foods and fasting. Muslim patients might use prophetic medicine, reciting verses, and dictates when affected by diseases, which is believed to synergise with conventional or CAM medicine usage. Both health products and Islamic religious precepts do not present conflicts or challenges to pharmacists, denoting an excellent health-related integration of the religious minority. Additionally, no significant impact is expected in pharmacy practice regarding substance interactions or the use of alternative therapies, such as cupping. Nonetheless, pharmacists should be aware of medication non-adherence, especially with some pharmaceutical ingredients and changes resulting from fasting. This situation is especially relevant in patients where medication schedules should be adjusted and conditions monitored.

Future studies should comprise a more detailed characterisation of medication-related behaviours for culturally and religiously diverse populations and the screening of medications available in the Portuguese market that might limit treatments in multicultural communities.

## Figures and Tables

**Figure 1 pharmacy-10-00055-f001:**
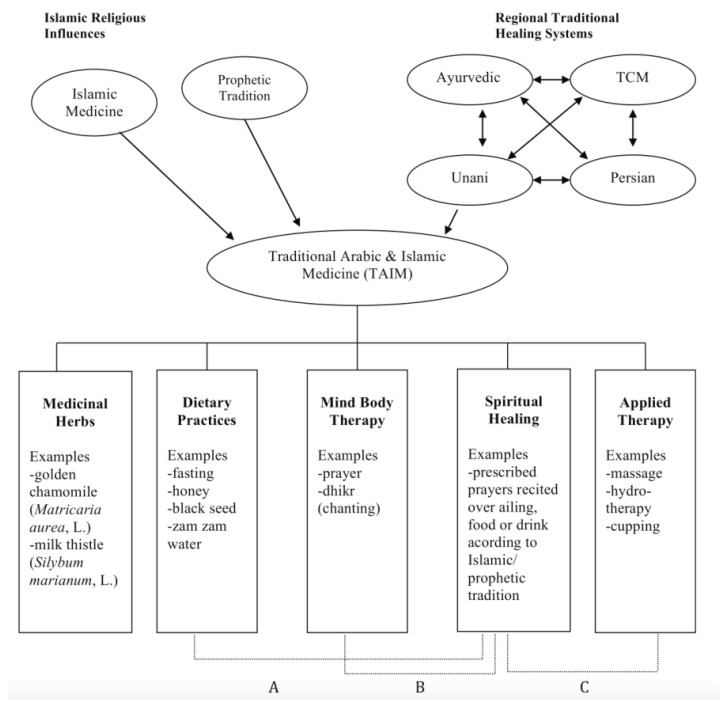
A conceptual model of Traditional Arabic and Islamic Medicine (source: Alrawi and Fetters, 2012). (**A**) represents dietary practices, (**B**) mind-body therapy, and (**C**) applied therapy.

**Table 1 pharmacy-10-00055-t001:** Chronic conditions and medications used (*n* = 37).

Chronic Diseases		ATC Drug Classification	
*Circulatory system (12%)*			
Arterial hypertension	5%	C09: Agents acting on the renin-angiotensin system	4%
		C07: Beta-blocking agents	5%
Hypercholesterolemia	4%	C1: Lipid modifying agents	3%
Hypertriglyceridemia	1%		
Other circulatory system diseases	2%	B01: Anti-thrombotic agents	2%
*Respiratory system (9%)*			
Asthma	5%	S02B: Corticosteroids	2%
Sinusitis/rhinitis	4%	R06A: Antihistamines for systemic use	11%
*Digestive system (7%)*		A02: Drugs for gastric disorders	2%
		N02: Analgesics	15%
*Immune system diseases (7%)*		L04: Immunosuppressants	2%

**Table 2 pharmacy-10-00055-t002:** Health products and their most common medicinal use (*n* = 100).

Health Products		Frequency of Use (%)	Total *n*. of Different Uses	Most Frequent Medicinal Use
Medicinal plants	Black cumin (seeds)	20	3	Helps protect against diseases
	Milk thistle (fruits and seeds)	6	2	Helps the digestive system
	Golden chamomile (flowers)	4	2	Helps to relax
Functional foods	Honey	89	8	Anti-inflammatory properties
	Olive oil	87	7	Helps general well-being
	Zamzam water	58	4	Helps general well-being
	Dates	5	1	Helps general well-being
	Figs	1	1	Helps prevent constipation

## Data Availability

Not applicable.

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
