# Peer review of "Exploring Muslims’ Health-Related Behaviours in Portugal: Any Impact on Quotidian Community Pharmacy Practice?"

_pharmacy, 2022, doi:10.3390/pharmacy10030055_

Round 1
Reviewer 1 Report
The paper is well written and there are no issues with the design. The sample size is small and the English need minor editing. Some of the statements are unclear.
Author Response
Thank you for your comments.
Corrections were made accordingly and followed the other reviewers' comments and suggestions.
AC, on behalf of the authors.
Reviewer 2 Report
Please check the attached document to address my comments and grammar errors. Overall, the research topic was very interesting since it was one of few studies that have evaluated how Muslim religion influences community pharmacy practice. I also liked the conceptual model presented in figure 1 which depicts how certain determinants affect patients and the use of CAM and prayer. The only other suggestion is also trying to provide the stats or case reports regarding how the lack of religious and cultural understanding for Muslim patients and the use of CAM had a negative influence on community pharmacy practice in the introduction (e.g., clinicians without cultural competency and empathy toward the traditional Hmong lifestyle or Hmong culture ended up reporting parents to the police for child abuse by not administering prescribed anticonvulsant to daughter). Also, try to list the relatively young (median age 28 yrs) study population as a limitation since the younger generation can be more westernized compared to older generations and thus could have influenced the study outcome. Even based on my own pharmacy practice experience, usually older patients tend to depend more heavily on CAM, traditional medicine, and prayers than younger patients. Other than that, this manuscript was beautifully written which addresses unusual topics in pharmacy practice. So really great job.

Author Response
Dear Reviewer,
Thank you for your kind reply. All the comments and suggestions were incorporated, as suggested.
AC, on behalf of the authors.

Reviewer 3 Report
Dear authors, thank you for letting me review your very interesting paper about Muslim health-related behaviour in Portugal. I am certain that this can give an insight to practising pharmacists and support them when counselling people of Muslim origin.
I have no comments on the introduction - thanks.
The study does however suffer from a very targeted and narrow data collection. The data was collected from a central mosque in Lisbon where educated professionals attend. Also, the role and involvement of Lisbon Islamic community leaders need to be described as this may have biased the data.
Materials and Methods:
You generated the survey with no reference to any other surveys. Is that because no similar approach exists?
When doing this ab ovo how did you approach this?
Did you submit the questionnaire? I can not find it?
Line 163-165. When doing the pilot, what method did you use to evaluate the open-ended questions and to investigate if the questions satisfied their intended purpose? Please add information
Line 181-185. Information is needed to know how open-ended questions were coded and validated.
Results and discussion
Please included reflections on the median age of participants - is this representative of Muslims in Portugal?
Also, please reflect on the statements about education line 273-274 + marriage line 276-278. With a median age of 27,5 years, I would expect participants to be better educated than the average and also a low rate of divorces as they only have been married a few years.
Strengths and limitations
Line 352-355. You state that this is one of the first studies to address practical aspects of Muslim patients' medication use etc. Please refer and include references to any previous work.
Line 358-359. What was the reason not to extend the questionnaire participation?
Please add a statement about the very low age of the included participants and how this significantly affects the representation of Muslims in Portugal and affects the quality of the study.
Conclusion
I acknowledge that the study (introduction) adds knowledge to community pharmacists. I am not sure how much the included data add to this knowledge as very few participants had any chronic health care problems.
Author Response
The authors would like to thank the reviewer for the comments and suggestions made, addressed according to the attachment.
AC, on behalf of the authors.

Round 2
Reviewer 3 Report
Dear Authors
Thank you for your reply and amendments. I appreciate you have considered all comments.
I do however still miss transparency about the process for the face validity and content analysis. Please insert a reference for the content analysis. Also, describe in more detail the process of getting answers to questions, generating an MS Excel file and using SPSS for analysis? E.g. quantification of open-ended questions.
I appreciate the limitations of this being the result of an MSc project, this is however not an argument when it comes to the publication of data.
This manuscript will certainly be of value to readers with limited knowledge of muslims health-related behaviour. My comments and suggestions are only to improve the quality of the manuscript.
Best regards,
Bjarke Abrahamsen
Author Response
The authors thank the reviewer for the input provided. Please find the authors' reply to the question below.
Q. Please insert a reference for the content analysis. Also, describe in more detail the process of getting answers to questions, generating an MS Excel file and using SPSS for analysis? E.g. quantification of open-ended questions.
R. A bibliography reference from a textbook on survey research methods (Sage encyclopedia) was introduced (reference 39). This is a well-cited reference from a prestigious publisher. According to pages 140-141, "... content analysis is a research method that is applied to the verbatim responses given to open-ended questions in order to code those answers into a meaningful set of categories
that lend themselves to further quantitative statistical analysis. In the words of Bernard Berelson, one of the early scholars explaining this method, ‘‘Content analysis is a research technique for the objective, systematic, and quantitative description of the manifest content of communication.’’ By coding these verbatim responses into a relatively small set of meaningful categories, survey researchers can create new variables in their survey data sets to use in their analyses." As presented in the textbook, the authors may describe the exact procedure from open responses to code frequencies in an SPSS file (the MS Excel file being just an intermediate data registry with no analytical procedures). However, this detail seems to belong to a methodological paper, while the technique is well known to most social researchers. Any reader wanting to know more about the procedure can access the example provided in the pages of the cited textbook.
In any case, the passage now reads: "and verbatim replies were content analysed, coding those textual answers into meaningful categories for further quantitative analysis of dichotomous variables [39]. Coding doubts were triangulated among the research team members,..."
Thank you.